

# Development and Testing of a Passive Sampler for Measurement of Gaseous Mercury

Ingvar Wängberg[1], Ulla Hageström[1], Jonas Sommar[2], Martin Ferm[1]

[1]IVL Swedish Environmental Research Institute, Gothenburg, 41133, Sweden
[2]Institute of Geochemistry, Chinese Academy of Science (IGAS), 99 Lincheng West Road, Guanshanhu District, Guiyang, Guizhou Province 550081, P.R.China

*Correspondence to*: Ingvar Wängberg (ingvar.wangberg@ivl.se)

**Abstract.** A passive sampler for measurement of gaseous mercury has been developed and tested in laboratory experiments. The sampler has also been compared to active measurement during field campaigns at ambient background concentrations of mercury as well as at slightly elevated concentrations. The result compares favourably with Tekran measurements and show that the sampler can be used for monitoring low background concentrations of mercury as well as elevated concentrations. It is also suitable as a personal dosimeter in occupational health measurements. The present work was performed as a subtask within the EU funded project, Global Mercury Observation System (GMOS).

Keywords: GMOS, Diffusive sampler, Total Gaseous Mercury (TGM), Background mercury concentration

## 1 Introduction

Mercury (Hg) is a poisonous heavy metal which occurs in the Earth's crust at low concentrations, primarily in the form of the mineral cinnabar (HgS, i.e. mercury sulphide). The metal is unique in many ways as it has a lower melting point (-39°C) and a lower boiling point (357°C) than any other metal. This means that mercury is volatile at room temperature and can occur in gaseous form, i.e. as Hg atoms ($Hg^0$), in the atmosphere. Other than the noble gases, mercury is the only element which occurs as an atomic gas at normal temperatures. Airborne mercury largely consists of atomic mercury or as it also is denoted Gaseous Elemental Mercury (GEM). Mercury in the atmosphere constitutes to more than 98 % of GEM, and small quantities of Gaseous Oxidised Mercury (GOM), as well as mercury bound to particulate matter. Mercury atoms are relatively stable in the atmosphere and have a residence time of about one year. That is long enough to allow transport on hemispherical scales before being oxidised and leaving the atmosphere via wet or dry deposition. Oxidised mercury ($Hg^{2+}$) can be converted to methylmercury in humid and oxygen-depleted environments via natural biological processes. Methylmercury is the most toxic form of mercury and is readily taken up and enriched in living organisms and hence possess a severe threat to the environment. Most of the mercury in the atmosphere is of anthropogenic origin. Artisanal gold mining, or small-scale mining, using mercury has recently been identified as the present greatest source of manmade emissions of



mercury to the atmosphere. This source is estimated to contribute to about 37 % of the total. Other important sources are coal combustion (24 %), primary non-ferrous metal production (10 %), cement production (9%) and large scale gold production, disposal of waste from mercury-containing products as well as emissions from contaminated sites etc. (AMAP report 2103).

Within the UNEP (United Nations Environment Programme), important efforts are being made to reduce global emissions of mercury. The Minamata Convention is an international agreement to reduce all use and emissions of mercury by man in order to protect humans and the environment (UNEP 2013). By March 2015, 128 countries had signed the convention and ten countries, including the USA, had ratified, i.e. undertaken the international agreement by implementation in their national legislation.


In the view  of the Minamata Convention the need of monitoring background concentrations of atmospheric mercury has increased (McLagan et al. 1016). Measurement of TGM using currently available automatic instruments is quite expensive and requires a constant electrical supply as well as trained instrument operators (Gustin and Jaffe, 2010). Hence, deployment of passive samplers for mercury measurement provides an economical alternative for a number of specific sampling needs

(McLagan et al. 1016).

Several papers on development and testing of diffusive samplers for airborne mercury are found in the literature (Kvietkus and Sakalys 1994; Brumbaugh et al., 2000; Skov et al., 2007; Gustin et al., 2011). Here a sensitive passive mercury sampler

developed within the EU project, Global Mercury Observation System GMOS, is presented. The passive sampler was first developed for measurement of elevated mercury concentrations, i.e. Total Gaseous Mercury (TGM) at contaminated sites, but has further been improved for measurement of ambient background TGM concentrations. The result from parallel passive-active measurements in Sweden and China are presented and discussed.


## 2 Experimental section

The performance of the passive samplers was compared with Tekran measurements by running Parallel passive-Tekran measurements at the EMEP measurement site Råö, during 2015-02-13 to 2015-05-25. Råö is situated on the West Coast of

Sweden ($57^o$ 23' 38''N, $11^o$ 54' 51''E, 7 m above sea level) and is one of the Master sites within GMOS where mercury species in background air continuously were measured using the Tekran 1130/1135 speciation system together with a Tekran Model 2537B Ambient Mercury Vapor Analyzer as the mercury detector. With the Tekran measurement system GEM rather





than TGM values are obtained. However, the difference between GEM and TGM is negligible at the Råö site and hereafter TGM and GEM values are considered to be equal.

Parallel passive-Tekran measurements were also performed in China in cooperation with the Institute of Geochemistry,
Chinese Academy of Science (IG-CAS). The measurements were performed at the roof of the chemistry building at IG-CAS during the period 2015-09-21 to 2015-10-21. The passive sampler result was compared to that of a new Tekran Model 2537X Ambient Mercury Vapor Analyzer. IG-CAS is situated in the new city of Guiyang, Jinyang. Gyiyang is the capital of the Guizhou province in south west of China and the location of IG-CAS is $26^o$ 39' 36''N $106^o$ 36' 37''E, 1288 m above sea level. The Jinyang city is surrounded by three coal-fired power plants, two cement production plants and an aluminium
smelter. Each measurement run lasted for 5 to 10 days. The intention with performing measurements at IG-CAS was to test the passive sampler at somewhat elevated mercury concentrations.

The passive sampler consists of a disk of 25 mm diameter and 12 mm thickness. Gaseous mercury is diffused into the device via a gas permeable membrane and is adsorbed onto a thin adsorbent based on active carbon. When not exposed the mercury
sampler is stored in an airtight plastic container as shown in Figure 1. After exposure the mercury trapped in the sampler is analysed by acid digestion and detection with CVAAS after reduction with tin(II) chloride and purging and trap on gold cartridges following the principle of double amalgamation. The TGM concentration is calculated as ng Hg per $m^3$ at STP (101325 Pa and 273.15 K). That is the same standardized concentration as is obtained with Tekran instruments.

The present passive sampling system has been utilised in a series of different passive samplers at IVL, such as samplers for $NO_2$, $SO_2$, and $O_3$ etc. and found to yield sensitive and accurate results. Due to the permeable membrane its influence on wind speed is low. Concentrations are calculated as a function of geometry, the mercury amount collected, time of exposure and temperature by applying Fick's first law. (Ferm 2001). Concentration at STP means that there is no influence from variation in atmospheric pressure (Ferm 2001). The influence from temperature is small; however, the average temperature
during each measurement period is used to correct the TGM uptake in respect to temperature. The mercury uptake capacity allows measuring concentrations in the range from ambient background concentrations to 100 μg m$^{-3}$.

At the measurements in Sweden and China four parallel passive samplers were deployed and compared to Tekran measurements.


Result from a laboratory study performed at IVL is also presented. In the experiments an airflow containing a constant concentration of mercury was generated using a thermostatted mercury source and circulated in a tubular exposure chamber. The experimental setup is schematically illustrated in Figure 2.



By help of a Mass Flow Controller (MFC) synthetic air was fed through a thermostatted Hg-source. The Hg-source contained a mercury droplet. Vapour from the mercury surface diffused through a narrow glass tube and mixed into a constant air flow. With the aid from a second MFC an air stream was fed through a humidifier, i.e. a gas wash bottle with de-ionised water. The two airflows were mixed before entering into the exposure chamber. The exposure chamber was kept at

room temperature (23 – 24 °C) and was equipped with two internal fans creating an air flow speed of 2 m s$^{-1}$. With the system the concentration in the chamber could be kept fairly constant and was measured by drawing 50 - 200 ml air volumes through an Au-trap, which then were analysed by means of thermal desorption and CVAF spectrometric detection of mercury. The average mercury concentration during each measurement was obtained by drawing sample air though an active carbon trap at a constant flow rate using MFC 3. The mercury collected with the active carbon traps were analysed by the

same method as described for the passive samplers above. In each experiment 2 – 4 passive samplers were exposed in the chamber. The mercury concentration was varied by altering the air mass flow rates through MFC 1 and 2 and an Hg concentration range of 0.2 – 2.5 μg m$^{-3}$ was obtained. Alteration of the flow rate also changed the humidity in the exposure chamber and due to that the relative humidity was varied from 50 to 82 % during the experiments.


## 3 Results and discussion

Eight measurements runs were accomplished at the Råö site in Sweden and the result is shown in Figure 3. In each run 2 to 4 passive sampler were measured in parallel and compared to a single Tekran measurement. Sample runs 2 and 4 were

performed during 31 and 28 days, respectively whereas the rest were of 14 days duration. The variation in mercury concentrations during the measurements was very small and according to the Tekran measurements confined to 1.42 – 1.60 ng m$^{-3}$. This corresponds to typical concentrations of mercury at the Råö site. Blue bars show average passive sampling results. Error bars indicate 1 standard deviation and ranged from 0.04 to 0.21 ng m$^{-3}$ with an average value of 0.14 ng m$^{-3}$, which corresponds to an average uncertainty of ± 9 %.


Four measurement runs were performed in Jinyan in China each including 4 passive samplers. The result is presented in Figure 4. Due to the proximity to mercury emission sources the variation in TGM was much higher at this site. Five minute average TGM concentrations varied between 1.98 to 39.3 ng m$^{-3}$ as can be compared to 1.10 to 2.24 ng m$^{-3}$ at the Swedish site. The average TGM concentration at Jinyang was 6.4 ng m$^{-3}$. The standard deviation of the passives samples ranged from

0.28 to 0.94 ng m$^{-3}$ which corresponds to an average uncertainty of ± 9 %.

Based on 3 times the standard deviation of field blanks, from 6 measurements, the detection limit regarding the Swedish measurements was calculated to be 0.9 ng m$^{-3}$ for a sampling period of two weeks. The field blanks from the Chinese measurements were somewhat higher but the standard deviation of field blanks (four in total) was less than those from the





Swedish site and a detection limit of 1 ng m$^{-3}$ was obtained for weekly measurements. The reproducibility of the mercury analysis was tested by performing double analysis which yielded an average reproducibility of 2 %.

The average passive TGM values from the Swedish and the Chinese sites were plotted against the Tekran values and the result is shown Figure 5. The measurement values form a fairly straight line with a slope close to one and with a close to zero intercept. The result indicates that although the scatter in individual passive TGM values is rather large average values compares favourably with the Tekran measurement measurements.

Results from the laboratory tests are shown in Figure 6. The two methods used to determine the mercury concentration in the
exposure chamber yielded almost identical results. GEM concentrations obtained in the experiments fits excellent with the Au-trap measurements, with a close to unity slope and low intercept. The variation among the passive samples were from 0.01 to 0.27 μg m$^{-3}$ with an average value of 0.09 μg m$^{-3}$ calculated as one standard deviation. This yields an average uncertainty of ±9 %. Hence, the same uncertainty as obtained in the other measurements.

**4 Conclusions**

As stated above the uncertainty of a single passive measurement is on average ± 9 % based of 1 standard deviation. If applying a cover factor of 2, the uncertainty can be expressed as ± 0.18 % at a confidence level of 95 %. With a background concentration of 1.5 ng m$^{-3}$ this uncertainty corresponds to ± 0.27 ng m$^{-3}$ for a single 14 days long measurement. Gustin et al. (2011) stated that for a TGM passive system to be useful it should be able to resolve concentration differences of 0.1 ng m$^{-3}$,
have little interference as a function of environmental conditions, and be configured such that there is a low risk for inadvertent contamination. The first criterion is obviously not met with the present passive sampler. In addition, the passive samplers seem to give slightly higher TGM values than the Tekran instrument as can be seen in Figure 1 and 2. On average the passive samplers gave 4 % higher TGM values in comparison to the Tekran instruments during both the Swedish and the Chinese measurements.


However, despite this, the scatter appears to be random as is indicated by the good fit between average passive TGM values and Tekran measurements. At the Swedish site the difference between the average passive TGM and TGM from the Tekran measurement range from 0 – 0.22 ng m$^{-3}$ with an average of 0.09 ng m$^{-3}$. The diffusive sampler is relatively insensitive to wind conditions and as shown in Figure 4 it may also be deployed for measurements of elevated TGM concentrations at
contaminated areas. It may also be used as a personal dosimeter in occupational health measurements.

**5 Acknowledgment**

The financial support from The European Commission, FP7 (contract no. 26511) is gratefully acknowledged.





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

# Figures


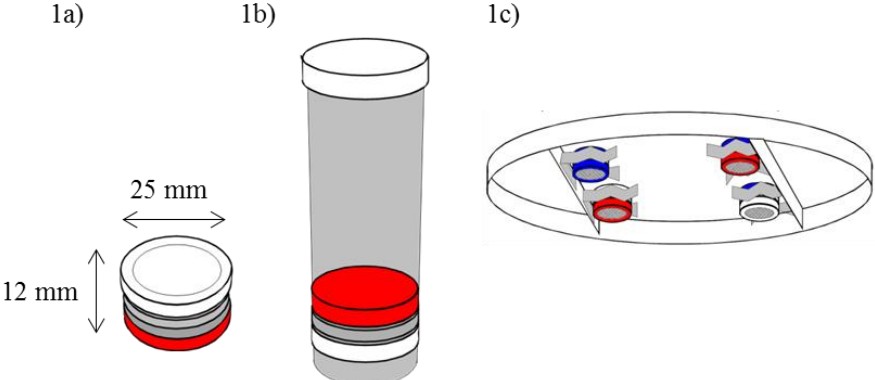

**Figure 1.** 1a) The Badge type passive sampler. 1b) The passive airtight sample storing container. 1c) The passive sample

5  holder-protective shield.

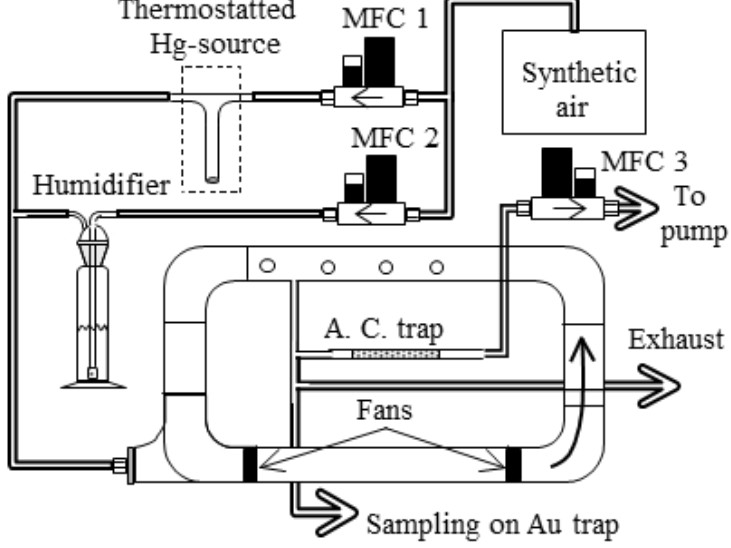

**Figure 2**. Tubular exposure chamber connected to a constant mercury flow system.

10

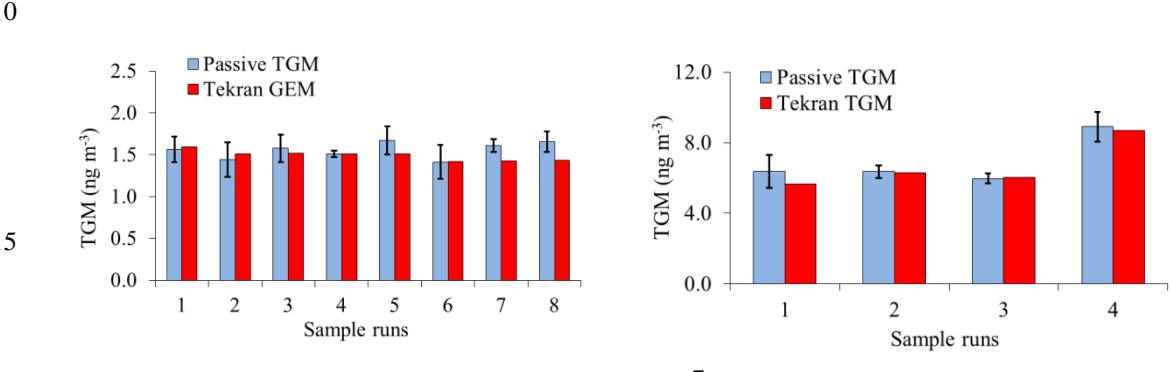

15



**Figure 3**. Råö measurements.    **Figure 4**. Jinyang measurements.

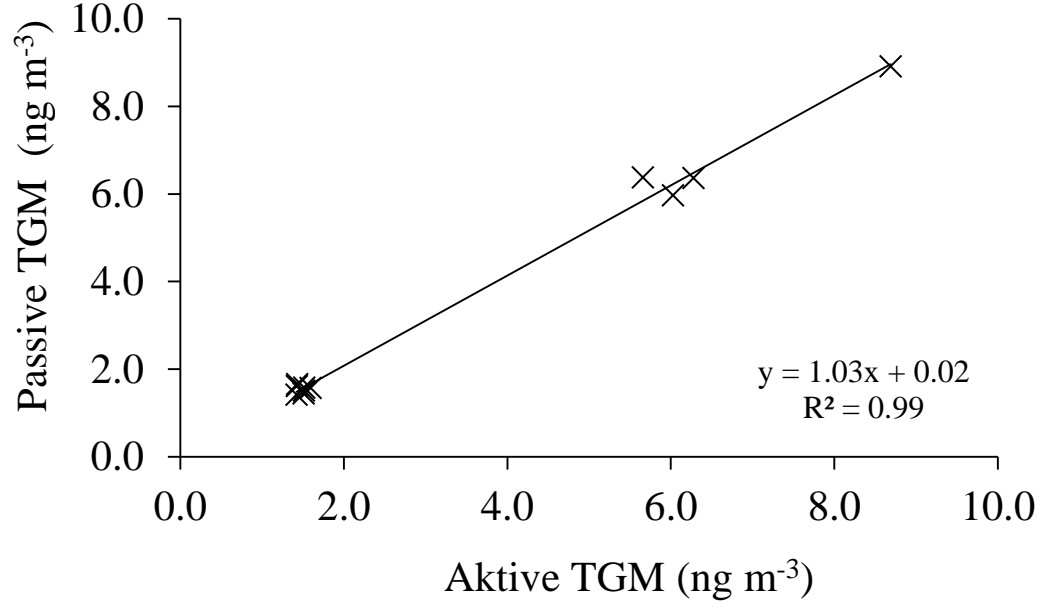

$$y = 1.03x + 0.02$$
$$R^2 = 0.99$$

**Figure 5**. Passive TGM sampling as function of active Tekran measurements at Råö in Sweden and Jinyang in China.

10





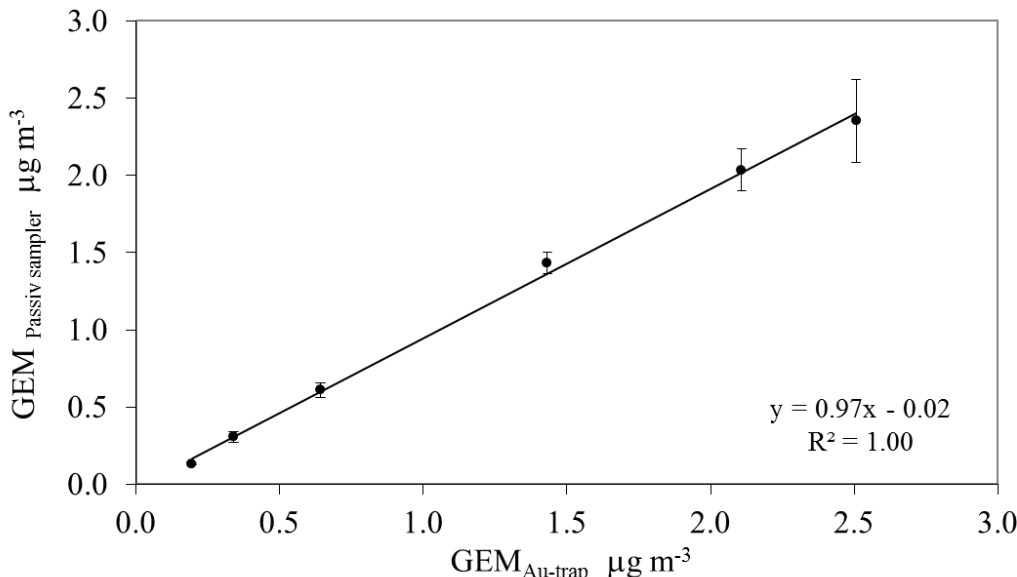

**Figure 6**. Result from the laboratory tests. Average passive GEM values are plotted against the measurements with the Au-traps. Error bars shows ± 1 standard deviation.

