# Peer review of "Development and Testing of a Passive Sampler for Measurement of Gaseous Mercury"

_Atmospheric Chemistry and Physics, 2016_

## Referee Comment (RC1) · Anonymous Referee #1 · 7 Jul 2016

This paper describes development of a passive sampler for Hg using activated carbon. The manuscript is a bit "sloppy" in terms of presentation and I have questions regarding the methods and results. There is no research hypothesis and tests have been inconsistently applied. Data from Figure 6 is not really valid for testing the passive sampler, because they use the passive sampler as an active sampler. There are no data to support the last statement in the abstract.

Introduction, first paragraph, "which" should be "that". This paragraph is poorly referenced. GOM can be much more than 2%. The word "severe" is a little extreme. AMAP ref date is 2103? 3rd paragraph and first sentence "of" should be "for". McLagan ref is 1016? How do they know the sampler is collecting only GEM? Experimental section First paragraph there are capitalization issues. Hg measured by the Tekran 2537/1130/1135 are operationally defined compounds not species. You do not know

whether the 2537 was measuring GEM or TGM see Gustin et al., 2013 EST. The exact configuration of the Tekran system at each location needs to be described. How long were samples collected at Rao? Second paragraph- why not a consistent number of days? Third paragraph- how is this instrument calibrated? How were the Tekrans calibrated? Fourth paragraph. Needs references for first sentence. There needs to be tests to demonstrate the influence of wind speed. It seems that these have not been adequately tested and there is speculation regarding temperature affect. Tests are needed. Last paragraph. This is not really a test of the passive system. Given the lack of tests and systematic measurements this work does not really advance science and there is no evidence based on the limited data that this can be used as a dosimeter.

---

## Referee Comment (RC2) · Anonymous Referee #2 · 12 Jul 2016

The manuscript sounds as in interesting study about the potentials of a kind of passive sampler used to adsorb elemental mercury from the atmosphere. Unfortunately the lack of data about the structure of the passive sampler, any description about the strategy of exposure, any time-relationship with Hg absorption, make the study unclear and unattractive from a scientific point of view, unless of providing more details.

a) Each detailed information reported in Introduction (p1, r20-r33, p2. R1-3) should be linked to proper literature reference. The cited link (p.2, r3 http://www.amap.no/) refers to a generic webpage.

b) Citations McLagan et al. 1016 should be properly changed.

c) A better description of the Hg passive sampler should be provided (r13-r14) together with the Figure 1 and the Figure caption, that are both very sparing of details.

d) Additional information should be provided about the Hg sampling (exposure), the shape and volume of the container.

e) (P5. R1-2). "The reproducibility of the mercury analysis was tested by performing double analysis which yielded an average reproducibility of 2 %". About such a sentence, would the authors better clarify what they mean when they talk about the reproducibility (of 2%) of the mercury analysis?

f) Final results should be more clearly described (p.5 r12-13). Authors never dealt with the relationship between the Hg adsorption and the time of exposure. Could they provide any information about this parameter?

g) What about the passive sampler saturation?

---

## Referee Comment (RC3) · Anonymous Referee #3 · 13 Jul 2016

Although the manuscript presents a measurement technique which is very promising for the ambient monitoring of gaseous mercury species, it lacks a robust description of the technique, details on the comparison with online measurements, a more robust analytical protocol and a deep discussion of the data.

Introduction The introduction needs more references Page 2 : TGM is not defined Passive sampler is not defined. What is the principle of this technique? What are the different passive samplers used in previous studies? Differences? Drawbacks?

Experimental section What is a Master site? "The difference between GEM and TGM is negligible at the Rao site": please provide some evidence. Page 3 line 6: the period does not match the 5-10 days measurement runs The passive sampler must be more described. The analytical technique for measuring Hg on the matrix as well. Why is

[Figure]

TGM measured? And not only GEM? Is there any testing? Calibration? There is a confusion between all the different experiments. There are 3 different experiments? What is the composition of synthetic air? How much Hg is there in it? Are the Tekran calibrated? Page 4, line 31: "field blanks": how are they obtained? Figure 2: lack of legend Figure 3: now you compare GEM vs TGM. Why is there no uncertainty on Tekran measurements?

Figure 5: This cannot be a robust comparison given the lack of data above 2 ng/m3 How can you affirm that the diffusive sampler is insensitive to wind conditions? I do not see any strong evidence. What is the influence of the temperature on long-term monitoring?

---

## Referee Comment (RC4) · Anonymous Referee #4 · 14 Jul 2016

The manuscript introduces a new passive sampler for gaseous mercury that is potentially a worthwhile contribution to this field. However, the paper has many serious shortcomings that would need to be addressed. The main ones are:

1. The description of the sampler and its deployment is insufficient.

2. The description of how volumetric air concentrations are derived from the amount of mercury quantified in the exposed sampler is insufficient.

3. The referencing is wholly inadequate.

In general, I think the paper (in much improved form) would be better suited for Atmospheric Measurement Techniques than for Atmospheric Chemistry and Physics.

The description of the sampler and its deployment is insufficient.

[Figure]

Presently a mere two sentences ("The passive sampler consists of a disk of 25 mm diameter and 12 mm thickness. Gaseous mercury is diffused into the device via a gas permeable membrane and is adsorbed onto a thin adsorbent based on active carbon") and Figure 1 are used to describe the sampler. This is woefully inadequate. The figure is insufficiently labeled.

- What is the "disk" made from? What is the commercial supplier of the "disk"? If it is not commercially sourced, how is it manufactured?

- What is the "gas permeable membrane" made from? What is the commercial supplier of this "membrane"? If it is not commercially sourced, how is it manufactured?

- What is the "thin adsorbent based on active carbon"? What is the commercial supplier of the "sorbent"? If it is not commercially sourced, how is it manufactured?

- Are the "membrane" and the "adsorbent" part of the "disk"?

- In Figure 1a, what is the red disk, what are the two grey disks, what is the white disk? What are the dimensions of each of these elements? The figure needs to be labeled.

- What is the diffusion path length? What is the porosity of the permeable membrane? What is it pore size?

The next sentence reads: "When not exposed the mercury sampler is stored in an airtight plastic container as shown in Figure 1."

- What plastic is the "container" made from? What is the commercial supplier?

- The Figure 1b is somewhat odd, as it does not really indicate that the sampler is inside the container, but rather attached to its bottom. Why is there so much air space in the container?

- How long can it be stored in this container? How should it be stored, e.g temperature?

The paper is entirely silent on how the sampler is being exposed to ambient air. Figure

1c appears to show a "passive sampler holder- protective shield", but is never referred to in the text and neither is it labeled.

- How is the sampler deployed? What is the shield made from? What are its dimensions? At what height above ground is it placed?

- In Figure 1c, what do they colors designate? What are the W-shaped grey bands?

- The lengths of deployments in Rao and Jinyan should be mentioned in the methods section. Presently, they can only be inferred from different places in section 3 ("Sample runs 2 and 4 were performed during 31 and 28 days, respectively whereas the rest were of 14 days duration." Presumably "weekly measurements" in Jinyan). The dates for each individual deployment and retrieval should be tabulated. It needs to be indicated which samplers were duplicated and which were done in quadruples.

The description of how volumetric air concentrations are derived from the amount of mercury quantified in the exposed sampler is insufficient.

In the passive sampler literature, the concept of a sampling rate is typically employed to convert the amount of analyte taken up by the sampler into a volumetric air concentration. Here the authors merely state: "The TGM concentration is calculated as ng Hg per m3 at STP (101325 Pa and 273.15 K)" without an explanation of how this is done. No sampling rate is given.

The following paragraph contains the sentence: "Concentrations are calculated as a function of geometry, the mercury amount collected, time of exposure and temperature by applying Fick's first law. (Ferm 2001)." This suggests that the sampling rate is theoretically estimated assuming molecular diffusion. Much more detail is required here: What is the assumed effective diffusion path length and how was it derived? How is the molecular diffusivity of mercury parameterized? The reference Ferm (2001) is insufficient as it is not readily accessible.

Why has it not been attempted to experimentally determine the sampling rate? Is it

not conceivable that the 4 % consistent discrepancy between Tekran data and passive sampler data is due to a theoretical sampling rate that is underestimated by 4 %? In fact, why not use the laboratory study results (displayed in Figure 6) to derive empirical sampling rates and use those to corroborate the theoretically derived ones?

The referencing is wholly inadequate.

The referencing is very poor:

- The long, first paragraph of the introduction includes many statements and assertions and yet only a single reference to an AMAP report.

- The introduction quotes four papers that had previously presented passive sampler for gaseous mercury. Why those and not others? In particular, why is there no reference to previous passive samplers based on active carbon as a sorbent (Zhang et al. 2012, McLagan et al. 2016)? How is the new sampler an improvement over those samplers?

- The following statement: "The present passive sampling system has been utilised in a series of different passive samplers at IVL, such as samplers for NO2, SO2, and O3 etc. and found to yield sensitive and accurate results." cries out for a reference.

- As stated before, a conference proceeding (Ferm, 2001) is an insufficient reference for a crucial piece of information required for a full understanding of the study.

Other items:

- The abstract is very poor and should be completely rewritten. For example, instead of "a sampler has been developed" describe the characteristics of the sampler. Instead of writing "has been tested" describe the tests and their results. The second to last sentence refers to information that is not addressed in the paper at all. The last sentence contains information that has no place in an abstract.

- Page 1, line 30: replace: "possess" with "constitutes"

- Page 2, line 8: Delete: ", i.e. undertaken"

- Page 2, line 31 and page 3, line 7: More information on the active TEKRAN measurements needs to be provided. QA/QC information, for example.

- Page 2, line 21: "its influence on wind speed is low". What does "its" refer to? Or should it rather say: "the influence of wind speed on the sampler's uptake kinetics is low". If that is the case, how do you know? Is there experimental evidence?

- Page 2, line 25: The statement on the uptake capacity is meaningless without specifying the length of deployment.

- Page 5, line 8: compare favorably (delete "s"). Also delete one of the two "measurements".

- Page 5, line 16: "based on 1 standard deviation"?

- Page 5, line 17: What is a "cover factor"?

- Page 5, line 30: Replace "health" with "exposure". What is the basis for making this statement: "It may also be used as a personal dosimeter in occupational health measurements"? Has it been tested accordingly?

- Either Figure 5 or Figure 3 and 4 are redundant as they show the same data. I suggest to delete Figs. 3 and 4 and include the standard deviation of the replicated passive sampler measurement in Figure 5. Also, the active TGM measurement of the TEKRAN surely has an uncertainty and should be estimated and displayed in the Figure(s).

- The figure captions are too short and not descriptive enough.

References

Zhang, W., Y. Tong, D. Hu, L. Ou, X. Wang. Characterization of atmospheric mercury concentrations along an urban–rural gradient using a newly developed passive sampler. Atmos. Environ. 2012, 47, 26-32.

McLagan, D.S., C.P.J. Mitchell, H. Huang, Y.D. Lei, A.S. Cole, A. Steffen, H. Hung, F. Wania. A high precision passive air sampler for gaseous mercury. Environ. Sci. Technol. Lett. 2016, 3, 24-29